# AutoSync: Learning to Synchronize for Data-Parallel Distributed Deep Learning

[1,2]**Hao Zhang**[†], [1,3]**Yuan Li**[†], [4]**Zhijie Deng**, [1]**Xiaodan Liang**,
[3]**Lawrence Carin**, [1,2]**Eric P. Xing**
[1]Petuum Inc., [2]Carnegie Mellon University, [3]Duke University, [4]Tsinghua University
{hao.zhang,christy.li,xiaodan.liang,eric.xing}@petuum.com,
dzj17@mails.tsinghua.edu.cn, lcarin@duke.edu

## Abstract

Synchronization is a key step in data-parallel distributed machine learning (ML). Different synchronization systems and strategies perform differently, and to achieve optimal parallel training throughput requires synchronization strategies that adapt to model structures and cluster configurations. Existing synchronization systems often only consider a single or a few synchronization aspects, and the burden of deciding the *right* synchronization strategy is then placed on the ML practitioners, who may lack the required expertise. In this paper, we develop a model- and resource-dependent representation for synchronization, which unifies multiple synchronization aspects ranging from architecture, message partitioning, placement scheme, to communication topology. Based on this representation, we build an end-to-end pipeline, *AutoSync*, to automatically optimize synchronization strategies given model structures and resource specifications, lowering the bar for data-parallel distributed ML. By learning from low-shot data collected in only 200 trial runs, AutoSync can discover synchronization strategies up to 1.6x better than manually optimized ones. We develop transfer-learning mechanisms to further reduce the auto-optimization cost – the simulators can transfer among similar model architectures, among similar cluster configurations, or both. We also present a dataset that contains nearly 10000 strategy and run-time pairs on a diverse set of models and cluster specifications.

## 1 Introduction

Recent advances in deep learning (DL) have benefited greatly from training larger models on larger datasets. To cope with the associated computational complexity, data-parallel training [19, 5, 37] has been introduced and reported remarkable successes in scaling up model training to thousands of GPUs [10] with billions of parameters [7, 28]. Data parallelism partitions and dispatches large datasets to multiple worker devices, derives gradients for each worker on its independent data split, and synchronizes gradients of all workers at the end of each iteration.

A variety of strategies and system implementations have been developed to facilitate *synchronization* in data-parallel training, such as systems specialized in different communication architectures [27, 37, 29, 5], message encoding methods [21, 35], or parameter partitioning or merging schemes [18, 16]. Achieving desired data-parallel speedup, however, requires the synchronization strategy to fit well with the statistical and algorithmic properties of the model and the cluster specification. For examples, bipartite parameter servers work well for models whose sparsity structure creates "hot spots" [4, 19, 18], while collective all-reduce outperforms other communication architectures when the majority of the distributed communication happens between GPUs [29, 10]. Existing systems struggle to provide

---

[†] Equal contributions. The work is done while Yuan Li was an intern at Petuum Inc.

excellent all-round performance on diverse models due to their oversimplified assumptions about the synchronization, and rigid application of fix-formed synchronization strategies (*e.g.,* parameter server (PS) [19, 33] for BytePS [27], Allreduce for Horovod [29]), ignoring the characteristics of models or clusters. More importantly, different strategies often exhibit sharp performance differences when applied to different ML building blocks (shallow, deep, sparse, dense, etc.) [37, 18], and the burden of selecting the *right* strategy for the model of interest is placed on ML practitioners, who may not have domain expertise on the trade-offs among these systems. Given the combinatorial number of choices for various synchronization factors, (*e.g.,* architecture, variable partitioning, and placement configuration), it is prohibitively costly to manually search for the optimal strategy, and the search has to be redone every time a new model is developed.

To address these challenges, this paper aims to answer: Can one automate the selection of the optimal synchronization strategy, given a model and cluster specification? To this end, we identify multiple synchronization-affecting factors in data-parallel distributed DL. By factorizing the strategy with respect to each trainable building block of a DL model, we construct a valid and large strategy space spanned by multiple factors. To efficiently navigate the space and locate the optimal strategy, we build an end-to-end pipeline, *AutoSync*. AutoSync leverages domain knowledge about synchronization systems to reduce the search space, and is equipped with a *domain adaptive simulator*, which combines principled communication modeling and data-driven ML models, to estimate the runtime of strategy proposals without launching real distributed execution. To further reduce practical development cost, we study the transferability of trained simulators across different models and resource specifications, which shows promising adaptability to unseen models or cluster configurations.

We evaluate AutoSync on a broad set of models and clusters, and show that there exist ample strategies in the proposed space that outperform hand-optimized systems by a significant margin. AutoSync can effectively find strategies that reduce the training time by 1.2x - 1.6x than hand-optimized ones on multiple, difficult-to-parallelize model architectures (e.g. NCF [13], BERT [7] and VGG16 [30]), within an acceptable budget. Leveraging transfer learning, AutoSync simulators can be trained on cheaper trial data collected on smaller models or clusters, and used to derive strategies without additional training for larger models or costlier clusters. As an additional contribution, we collect a dataset containing nearly 10000 data points containing (model, resource, strategy) tuples and their corresponding runtime on real clusters. We share the dataset with the community to encourage extended studies.[1]

## 2  Problem Definition

**Background.** We represent a DL model using its dataflow graph $\mathcal{G} = \{(V_{\mathcal{G},\theta}, V_{\mathcal{G},o}), E_{\mathcal{G}}\}$ where $V_{\mathcal{G}}$ are nodes in $\mathcal{G}$ including trainable variables $V_{\mathcal{G},\theta} = \{v_i\}_{i=1}^{|V_{\mathcal{G},\theta}|}$ or computational operations $V_{\mathcal{G},o}$, and $E_{\mathcal{G}}$ are tensors (edges) transferred between nodes. For simplicity, we use $V$ equivalently with $V_{\mathcal{G},\theta}$ to notate the set of variables. In addition to $\mathcal{G}$, we define a cluster as a device graph $\mathcal{D} = \{V_{\mathcal{D}}, E_{\mathcal{D}}\}$, where $V_{\mathcal{D}} = \{d_p\}_{p=1}^{|V_{\mathcal{D}}|}$ represents devices (e.g. CPUs or GPUs), and $E_{\mathcal{D}} = \{b_{i,j}\}$ is a symmetric matrix with the entry $b_{i,j}$ representing the connectivity (e.g. bandwidth) between $d_i$ and $d_j$. In data-parallel training, we replicate $\mathcal{G}$ on all devices, and update each trainable variable $v_i$ using the aggregation of the stochastic gradients $\nabla_{v_i}(\mathcal{G}, X_p)$ computed by each worker device $d_p$ on its data partition $X_p$, following $v_i^{(t+1)} \leftarrow v_i^{(t)} + \epsilon \sum_{p=1}^{P} \nabla_{v_i^{(t)}}(\mathcal{G}, X_p)$, for $v_i \in V_{\mathcal{G},\theta}$. Since devices are distributed across the cluster, obtaining the aggregation requires *synchronization* support, which collects updates $\nabla_{v_i^{(t)}}$ and provides all devices the shared access to a consistent version of $v_i^{(t+1)}$. Existing systems aim to optimize some individual factor to expedite synchronization, ignoring that the optimal of each factor significantly changes with $\mathcal{G}$ and $\mathcal{D}$. For ML practitioners, it is challenging to select appropriate synchronization strategies for their models of interest without domain expertise.

**Problem formulation.** Alternatively, we pose the strategy selection as an optimization problem, in which the *per-iteration runtime* (e.g. time taken to process a batch on all nodes of the entire cluster, equivalent with system throughput), denoted as $f$, is minimized given $\mathcal{G}$ and $\mathcal{D}$ by solving

$$\min_{\mathcal{S}} f(\mathcal{G}, \mathcal{D}, \mathcal{S}), \text{ s.t. } \mathcal{C}, \tag{1}$$

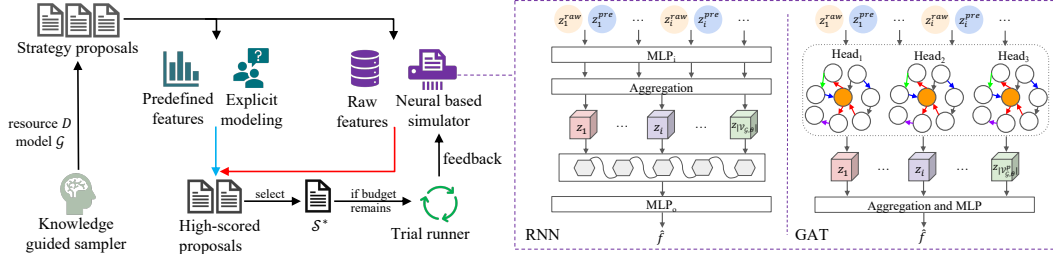

**Figure 1: Left**: Learning to synchronize framework. Initially, the simulator utilizes domain-agnostic features (Section 3.1) to explicitly estimate the runtime so to select promising strategies for evaluation (the blue line). After trials, the real runtime data are feedbacked to train the ML-based simulator to adapt to specific $\mathcal{G}, \mathcal{D}$, enhancing its capability in differentiating high-quality strategies. Gradually, the ML-based simulator takes over and directs the search (the red line). **Right**: Illustrations of the RNN and GAT simulators.

where we use $\mathcal{S}$ to denote a model and resource dependent representation of the synchronization strategy, and $\mathcal{C}$ as a set of constraints (developed in §3.2). Approaching this problem analytically needs continuous characterizations of $\mathcal{S}$ and $f$, which are unavailable. In light of the recent advance in AutoML [39, 36, 3], we define a domain-specific space considering multiple synchronization-affecting factors, and resort to search-based methods to find near-optimal $\mathcal{S}^*$.

**Search space.** When constructing the search space, we have the following considerations. First, instead of optimizing a single factor in a piecemeal fashion as commonly done in existing systems, we seek a unified space covering multiple synchronization-affecting factors, to capture the subtleties between them and their dynamics with different $\mathcal{G}$ and $\mathcal{D}$ via co-optimization. On the other hand, we want to establish direct correspondences between $\mathcal{S}$ and each participating variable of $\mathcal{G}$ in synchronization, so that the strategy can adapt with specific variable-wise mathematical properties.

Hence, we first decompose existing fixed-formed systems/strategies into following orthogonal factors[2]: (1) *Variable partitioning*, represented as $\boldsymbol{p}_i = [p_i^j]_{j=1}^{k_i}$ for the variable $v_i$, where $k_i$ is the number of tensor dimensions of $v_i$, and $p_i^j$ represents the *partition degree* on the $j$th axis. (2) *Variable placement*, defined as $\{d_i\}_{i=1}^m \subset V_{\mathcal{D}}$ which is the set of devices the variable resides. The placement being a single device means $v_i$ is shared across all devices. (3) *Synchronization architecture*: we define two types of architecture primitives, namely parameter server (PS) and collective communication (CC), and their architecture-specific semantics. In PS, we use *reduction hierarchy* to indicate whether parameters are transmitted hierarchically (*e.g.,* from a central CPU to multiple GPUs co-located on the same machine). In CC, we define *merge group*, where communication primitives assigned with the same group are merged and communicated via a single message, and *device order*, specifying the message passing order across devices (e.g. tree, ring). (4) *Message encoding and decoding*, notated as $c_i$ for $v_i$, introduces compression or decompression schemes to represent how messages are processed before (after) communication, enabling optimizations [34, 37, 21] that exploit structures (*e.g.,* low-rank) exhibited in the messages to reduce message size and fasten network transfer.

Based on these factors, we express the strategy as $\mathcal{S} = \{s_i\}_{i=1}^{|V|}$ where $s_i$, as a sub-strategy, includes the discrete choices of above factors for each $v_i \in \mathcal{G}$. Note that we decide whether to partition variables or not before any other aspects, so a sub-strategy needs to be generated for each variable partition. The multiple factors span a combinatorial space whose size grows with the size of $\mathcal{G}$ and $\mathcal{D}$. We next develop the *learning to synchronize* framework to approximate the optimal strategy $\mathcal{S}^*$.

## 3 Learning to Synchronize

Despite the large space, solving Eq. 1 poses an additional challenge: searching for $\mathcal{S}^*$ requires evaluating $f$ for each strategy proposal, which involves distributed execution on clusters and is prohibitively expensive. To make the search tractable, we present the learning to synchronize framework, illustrated in Figure 1, with two novel components: runtime simulation of arbitrary $\mathcal{S}, \mathcal{G}, \mathcal{D}$ (§3.1), and knowledge-guided strategy search (§3.2).

In detail, to reduce real execution, we develop a *domain adaptive simulator* to estimate $f$. The estimation is made possible (without training data) by first designing features agnostic to $\mathcal{G}, \mathcal{D}$. The features describe critical impacting factors on the runtime using predefined modeling, and can be generalized to any unseen $\mathcal{G}$ and $\mathcal{D}$. Then, we enhance them using ML models and various raw features extracted from specific $\mathcal{G}, \mathcal{D}$. In order to direct the search to the subspace where good strategies may locate, we instantiate constraints $\mathcal{C}$ in Eq. 1 using prior knowledge on synchronization systems. The final framework, named as *AutoSync*, approaches $\mathcal{S}^*$ alternately. In the initial phase, the simulator uses the training-free features to propose the initial batch of strategy candidates, which are executed on real clusters to obtain ground truth runtime. The low-shot data are feedbacked to train the simulator to adapt to $\mathcal{G}, \mathcal{D}$, achieving improved capability in differentiating high-performing strategies. The process alternates until the optimization budget for real distributed cluster evaluation is exhausted. We next describe the design of the simulator and the guided search.

## 3.1 Domain Adaptive Simulator

The simulator takes $(\mathcal{G}, \mathcal{D}, \mathcal{S})$ as input, and estimates its per-iteration runtime (equivalent with throughput). Because variable partitioning will alter $V_{\mathcal{G},\theta}$, we first infer the new set of variables $V'_{\mathcal{G},\theta}$ based on $\mathcal{G}, \mathcal{S}$, and let the simulator work with each variable $v_i \in V'_{\mathcal{G},\theta}$, which contains variable shards after partitioning original variables. We define a particular $(\mathcal{G}, \mathcal{D})$ target as a *domain*. When equipped with domain-agnostic features, the simulator is capable of estimating the runtime of any $(\mathcal{G}, \mathcal{D})$ without training. This is realized by systematic modeling of the runtime in distributed execution.

### 3.1.1 Predefined Modeling

We model the per-iteration runtime $T$ of parallelizing $\mathcal{G}$ on $\mathcal{D}$ using two contributing components: computation time $T_{\text{comp}}$, and parameter synchronization time $T_{\text{sync}}$. Since in data-parallel training one component usually dominates the other [11], we simply obtain $T$ via $T = \max(T_{\text{comp}}, T_{\text{sync}})$. We factorize $T_{\text{comp}}$ and $T_{\text{sync}}$ w.r.t. variables, similar to $\mathcal{S}$, and estimate $T_{\text{comp}}(v_i), T_{\text{sync}}(v_i)$ for each $v_i$ based on its $s_i$ in $\mathcal{S}$. $T_{\text{comp}}(v_i)$ can be approximated by profiling its corresponding operation on a single-device. To calculate $T_{\text{sync}}(v_i)$, we split $V'_{\mathcal{G},\theta}$ into variables using PS as $V^{PS}$ and using collective communication as $V^{CC}$, and derive two analytic forms of $T_{sync}(v_i)$.

For $v_i \in V^{PS}$, if we denote the size of the message by $v_i$ as $c_i(m_i)$ where the encoding/decoding scheme $c_i \in s_i$ has been applied on the original size $m_i$, and $w_i$ as the number of worker nodes involved in synchronizing $v_i$, the communication time $T_{\text{server}}^{PS}$ on the server hosting $v_i$, indexed as $j$, is

$$T_{\text{server}}^{PS}(v_i) = \underbrace{\sum_{k=1}^{w_i} \mathbb{I}_d(j,k) \cdot \frac{c_i(m_i)}{b_{j,k}} \cdot r_{i,k}^{\mathbb{I}_p}}_{\text{network transfer}} + \underbrace{\sum_{k=1}^{w_i} \mathbb{I}_d(j,k) \cdot r_{i,k}^{\mathbb{I}_p} \cdot \phi}_{\text{network overhead}} + \underbrace{\delta}_{\text{GPU kernel latency}}, \tag{2}$$

where $r_{i,k}$ is the number of replicas (e.g. number of GPUs) of $\mathcal{G}$ on worker $k$, $\mathbb{I}_d(j,k)$ and $\mathbb{I}_p$ are true when server $j$ and worker $k$ locate on different machines and when hierarchical reduction is used, respectively. $\phi$ and $\delta$ are network overhead and GPU kernel latency constants. The 1st term corresponds to sending messages from each worker $k$ to the server $j$ (and vise versa). The 2nd term captures network overheads that scale linearly with the number of workers, or with the number of replicas when $\mathbb{I}_p$ is false. The 3rd term captures constant GPU memcpy latency.

In addition to the formula, we can construct domain-agnostic features of $v_i$ as $z_i^{PS} = $ [network transfer, coefficient of $\phi$, coefficient of $\delta$]. As synchronizing any $v_i$ between any pair of nodes can happen simultaneously and is upper bounded by the multi-flow bandwidth, the communication bottleneck may be caused by the slowest transmission, or the total time of transmissions. Thus, we define global features for estimating $T_{sync}^{PS}$ as $z^{PS} = \max\{z_i^{PS} \text{ for } v_i \in V^{PS}\} \oplus \sum\{z_i^{PS} \text{ for } v_i \in V^{PS}\}/|V^{PS}|$, where $\oplus$ is vector concatenation, and $\max, \sum$ are elementwise.

Similarly, for $v_i \in V^{CC}$, we can estimate its communication time, and construct the global features of $T_{sync}^{CC}$ as $z^{CC}$. Detailed formulas are in the supplementary material. Concatenating $z^{PS}$ and $z^{CC}$ obtains the set of domain-agnostic features $z^{pre}$ for $(\mathcal{G}, \mathcal{D}, \mathcal{S})$. We can either use the estimated $T$ to rank different $\mathcal{S}$, or use the constructed features as inputs to ML models, which we elaborate next.

### 3.1.2 Domain-specific Modeling

Once trial data are acquired, we augment $z^{pre}$ using raw features $z^{raw}$ extracted from $\mathcal{G}, \mathcal{D}, \mathcal{S}$, and train ML models so as to capture their domain-specific characteristics. For each $v_i$, $z_i^{raw}$ vectorizes attributes including variable placement, synchronization architecture, encoding/decoding type, and merge group from $\mathcal{S}$, bandwidth and the number of replica devices of each node from $\mathcal{D}$, and variable size, dimensions, the sparsity of gradients, data types, and information of partitioned shards from $\mathcal{G}$. Combining $z^{pre}$ and variable-specific raw features $z_i^{raw}$, we adopt three different ML models: (1) a linear model, (2) a recurrent neural network (RNN), and (3) a graph attention networks (GAT) [38], to learn from trial data and make more accurate predictions of $f$.

**Linear Model.** The linear model, simply written as $\hat{f} = z^{pre} \cdot \boldsymbol{\theta}$, introduces trainable weights $\boldsymbol{\theta}$ and predicts the runtime $\hat{f}$ using only global features $z^{pre}$.

**RNN.** We use an RNN to model different $\mathcal{G}$ with varying numbers of variables, so as to inject $z_i^{raw}$. The RNN first concatenates $z_i^{pre}$ with $z_i^{raw}$, and then transforms the combined features via an MLP. The results are aggregated into $|V_{\mathcal{G},\theta}|$ features corresponding to the variables in the $V_{\mathcal{G},\theta}$. A bidirectional LSTM scans them following the forward-backward propagation order preserved in $\mathcal{G}$. At last, the prediction $\hat{f}$ is obtained via an extra MLP (Figure 1, detailed formulas in the supplementary).

**GAT.** We bring in GAT to model the raw graph structure of $\mathcal{G}$. To do so, we prune $\mathcal{G}$ into a graph $\mathcal{G}^p = (V_{\mathcal{G},\theta}^p, E_{\mathcal{G}}^p)$ with only variable (including partitioned) nodes and edges connecting them. For each node, similar to RNN, its raw and predefined features are concatenated, and features of all nodes and corresponding edge information are fed to a GAT for encoding graph structure as: $\{z_i\} = \text{GAT}(\{z_i^{raw} \oplus z_i^{pre} \text{ for } v_i \in V_{\mathcal{G},\theta}^p\}, E_{\mathcal{G}}^p)$, shown in Figure 1. The node features are then aggregated into graph-level ones, followed by an MLP to get the runtime estimation $\hat{f}$.

### 3.1.3 Training Objective

Accurately predicting runtime can be challenging due to uncertain factors on a distributed cluster, we instead train the simulators with the pair-wise logistic ranking loss [1, 3]:

$$\sum_{i=1}^{n} \sum_{j=1}^{n} \mathbb{I}(f_i + \sigma \mathbb{I}_1(P_i, P_j) > f_j) \log(1 + \exp(\hat{f}_j - \hat{f}_i)) \tag{3}$$

where $f_i, f_j$ are ground truth runtime, $\mathbb{I}$ is an indicator function, and $n$ denotes the number of training examples. Note that we augment the original ranking loss with a penalty term $\sigma \mathbb{I}_1(P_i, P_j)$, where $\sigma$ is a non-negative threshold, $P_i$ is the total number of partitions in $\mathcal{S}_i$, and $\mathbb{I}_1(x, y)$ outputs 1 when $x > y$, -1 when $x < y$, and 0 otherwise. This term additionally considers inherent partitioning overheads, and alleviates the bias toward heavily partitioning variables.

## 3.2 Knowledge-guided Search

The search uses the simulator to evaluate a strategy, and proposes candidates with low predicted runtime $\hat{f}$. It then selects a small number of qualified candidates from a large set of proposals for trial execution, as shown in Figure 1. We implement two search algorithm variants: random search and genetic algorithm (GA) [6].

**Knowledge constraints.** As the strategy space is exponentially large, it is inefficient to grid search the entire space. We instantiate the constraints $\mathcal{C}$ in Eq. 1 using two system design principles to restrict the exploration within promising regions. (1) Load balancing constraint $c_{lb}$: which is vital to alleviate the communication bottleneck. When deciding the placement for each $v_i \in V^{PS}$, $c_{lb}$ enforces to sample the placement from a multinomial distribution over all participating nodes in $\mathcal{D}$, where each node's probability of being chosen correlates to its current communication load and maximum bandwidth, so that nodes with higher available bandwidth are more likely to be sampled. This allows generating randomized solutions while approximately maintaining a balanced status across all nodes. (2) Adjacent merging constraint $c_{am}$: the fusion of collection operations should correspond to the model forward-backward propagation order in $\mathcal{G}$, as merging two operations in the head and tail of the model would prevent low-level scheduling overlapping communication and computation. $c_{am}$ is introduced to ensure variables adjacent to each other in $\mathcal{G}$ are more likely to be grouped together. We show the $c_{lb}$ and $c_{am}$ empirically improve search efficiency and quality in §5.1.

**Select evaluation candidates.** Similar to AutoTVM [3], we select the final set of candidates, that have minimized weighted sum of $\hat{f}$ and internal similarity, for distributed execution. The similarity can be estimated by either comparing strategy representation or the hidden outputs of simulators. We defer the details of how to select candidates and how to estimate the similarity between strategies to the supplementary material.

### 3.3 Low-cost Optimization using Transfer Learning

Performing end-to-end strategy optimization from scratch for a target domain $\mathcal{G}_t, \mathcal{D}_t$ on large clusters or large models is costly, especially if $\mathcal{G}_t$ will be only trained once (one-shot training). For unseen $\mathcal{G}_t, \mathcal{D}_t$, we consider transferring simulators trained on the data from a source domain $\mathcal{G}_s, \mathcal{D}_s$. This would be advantageous if the source domain data already exist, or any of $\mathcal{G}_s$ and $\mathcal{D}_s$ is small so their trial run data is cheap to collect. The transferability is made possible by our feature design – the predefined features are universal due to its domain-agnostic nature. The per-variable raw features are extracted from $\mathcal{G}$, $\mathcal{S}$, and $\mathcal{D}$, where $\mathcal{G}$ is the generic low-level dataflow graph representation, capable of expressing all NN models; $\mathcal{S}$, based on $\mathcal{D}$, is generated from a strategy space invariant to models as well. Hence, the feature components of different $(\mathcal{G}, \mathcal{D}, \mathcal{S})_i$ are invariant, leaving the only variability as the the length of features, caused by the different number of variables in different $\mathcal{G}$, which, however, is absorbed by models like GAT and RNN that operate on inputs with variable lengths. §5.2 validates the effectiveness of the transferable feature representation.

## 4 Related Work

**ML for systems.** There is a surge of interest in applying ML to solve system problems. Mirhoseini al. [25, 24] develops reinforcement learning (RL) frameworks to decide the placement of nodes in dataflow graphs. Paliwal *et al.* [26] combines RL and genetic algorithms (GA) to minimize the execution cost of NN graphs for compilers. AutoTVM [2] builds an ML-based pipeline to generate operator implementations better than hand-designed. AutoSync belongs to this line of work: it uses ML to optimize data-parallel synchronization and addresses the unique challenges therein.

**Synchronization system autotuning.** Many data-parallel ML systems demonstrate certain levels of autotuning capability. Horovod [29] and ByteScheduler [27] introduce adjustable *knobs* and *credit size*, respectively, and use Bayesian optimization (BO) [8] to autotune their values. Parallax [18] develops a 3-parameter linear model, learned via trial data, to find the best partitioning for sparse variables in PS. This work focuses on autotuning one or two hyperparameters of a specific strategy; AutoSync contrasts them by co-optimizing a generic and holistic representation of synchronization. Among them, the closest to ours is AutoTVM [3]. We draw insights from AutoTVM, but address a fundamentally different problem – data-parallel auto-distribution of ML training on clusters – which requires the problem-specific design of strategy representations, features, runtime simulations, etc.

**Automatic ML parallelization.** Automating the parallelization of ML programs is an ultimate goal in distributed ML. FlexFlow [16, 17] proposes the SOAP representation to express partitioning schemes of NN layers, and an MCMC-based algorithm to search for optimal partitioning configurations. TOFU [32] concerns the partitioning to finer-grained computational operations in DL dataflow graphs. They intersect with Autosync as data parallelism (which AutoSync focuses on) can be equivalently represented as partitioning all the layers/operations along their *batch* dimension. However, beyond partitioning, AutoSync models many other synchronization-affecting factors, such as communicating architectures, sharding, and merging schemes, *etc.*, which differentiates us from this line of work.

## 5 Evaluation

**Implementations.** We generate strategies on top of TensorFlow 2.0, and resort to it for distributed execution [9]. We treat $f$ as *system performance*, and conduct synchronous training[3] on 10 models with standard settings suggested by MLPerf [22], including an enlarged dense (x16x32) version of the neural collaborative filtering (NCF) [13], Transformers [31] and BERT variants [7], and various CNNs [12, 30, 15]. We managed to train all models to the suggested accuracy, hence skip the comparisons on convergence.

**Experiment setup and baselines.** We conduct experiments on two clusters ($\mathcal{D}$): (1) *Cluster A* is an in-house cluster with 11 nodes, each equipped with a TITAN X GPU and 40GbE Ethernet switch; (2) *Cluster B* is based on AWS, consists of 4x g4dn.12xlarge nodes, each with 4 NVIDIA T4 GPUs and 50GbE full bandwidth. We introduce two strong hand-optimized synchronization systems as external baselines. (1) Horovod [29]: which uses *AllReduce* (*AllGather*) to synchronize dense (sparse) gradients of all model variables, and utilizes BO to autotune collective operation fusion. (2) PS: a manually optimized PS with multiple optimizations [20, 37, 18, 27] incorporated, such as maintaining load balance across servers by partitioning and placement [20], autotuning credit size [27], and reducing network transfer via hierarchical reduction, *etc.* We use them as collective- and PS-based baselines, respectively. Note that both methods use the same TF and NCCL version for distributed execution and communication with AutoSync, preventing variations caused by systems. Throughout our research, we have collected around 10,000 data points containing $(\mathcal{G}, \mathcal{D}, \mathcal{S})$ input tuples and their ground truth runtime on 11 models and 14 clusters. More details about the system implementations, model and cluster settings, baseline performance, and the public dataset are described in the supplementary material.

## 5.1 End-to-end Results and Ablation Studies

**Comparing model instantiations.** To compare the linear model, RNN and GAT, we construct datasets using trial data collected on 6 different settings, and train them as simulators, respectively. We report their ranking accuracy on held-out test sets in Table 1 (the standard deviation of the results is within $\pm 2\%$ across 3 runs). RNN, by leveraging raw features, outperforms the linear model mostly. GAT, though additionally modeling the graph structure of $\mathcal{G}$, does not demonstrate substantial advantages. We hypothesize that GAT might need more data for training, which are unavailable in our budgeted search. We hence use RNN by default in the rest of the paper.

| Setting | Linear | RNN | GAT |
|---|---|---|---|
| NCF-dense, A | 0.771 | **0.894** | 0.810 |
| NCF-dense, B | 0.826 | **0.913** | 0.830 |
| VGG-16, A | **0.868** | 0.796 | 0.753 |
| VGG-16, B | 0.833 | **0.839** | 0.692 |
| BERT-base, A | 0.758 | 0.746 | **0.775** |
| BERT-base, B | 0.807 | **0.867** | 0.760 |
| BERT-large, A | 0.780 | **0.847** | 0.771 |
| BERT-large, B | **0.796** | 0.755 | 0.784 |

**Table 1:** Comparisons of different model instantiations of the simulator.

**Search algorithm comparisons.** We compare two search algorithms random search (RS) and GA on optimizing strategies for (VGG-16, A) with a budget of 200 trials. Table 2 reports the statistics of the per-iteration time of the 200 strategies found. GA by nature maintains a better average quality and lower variance than RS, but tends to overfit with the simulator's (inaccurate) prediction, resulting in worse quality on the best found strategy. We therefore prefer RS as the default algorithm than GA for AutoSync strategy optimization.

**Auto-optimization results.** We use AutoSync to optimize the strategies of NCF-dense (122M), VGG16 (138M), and BERT-large (340M). They cover 3 different NN families but are all considered "difficult-to-parallelize" workloads because of having >100M parameters. We compare two AutoSync variants with external baselines: (1) AutoSync(-s) where the simulator is disabled for searching. It randomly explores 30K strategies and selects 200 candidates that have minimized similarity (§3.2). (2) AutoSync: the full AutoSync with the budget of real evaluation on clusters as 200. To obtain the runtime $f$, we run 10 warm-up iterations, then another 40 iterations

| Stats | RS | GA |
|---|---|---|
| Mean (s) | 1.07 | 0.67 |
| Std. (s) | 0.63 | 0.03 |
| Min. (s) | **0.60** | 0.64 |
| Max. (s) | 2.34 | 0.79 |

**Table 2:** The per-iteration time statistics of 200 strategies found by RS and GA on (VGG16, A).

of training, whose runtime is averaged as the groundtruth. Figure 2 compares the *best* found strategy in 200 trials by two variants with the two manually optimized baselines. In 4 out of 6 settings, AutoSync(-s), *without a simulator*, discovers strategies up to 1.4x faster than the best one in PS and Horovod. With the simulator, AutoSync finds strategies 1.2x to 1.6x faster than baselines. To interpret the speedup, practically BERT-large needs 2M steps [23] of training to its reported accuracy with batch size 128 (batch size 8 on 16 GPUs). A 1.2x speedup reduces the training time by 7 days, and saves approximately $2200 AWS credits *per training job* on Cluster B. Moreover, in practice a model needs to be retrained when being applied to unseen data, but a trained simulator can be repeatedly used across jobs. Comparing AutoSync to AutoSync(-s): besides higher quality, the simulator guides the search to solutions sooner, e.g. on (BERT-large, A), AutoSync locates strategies

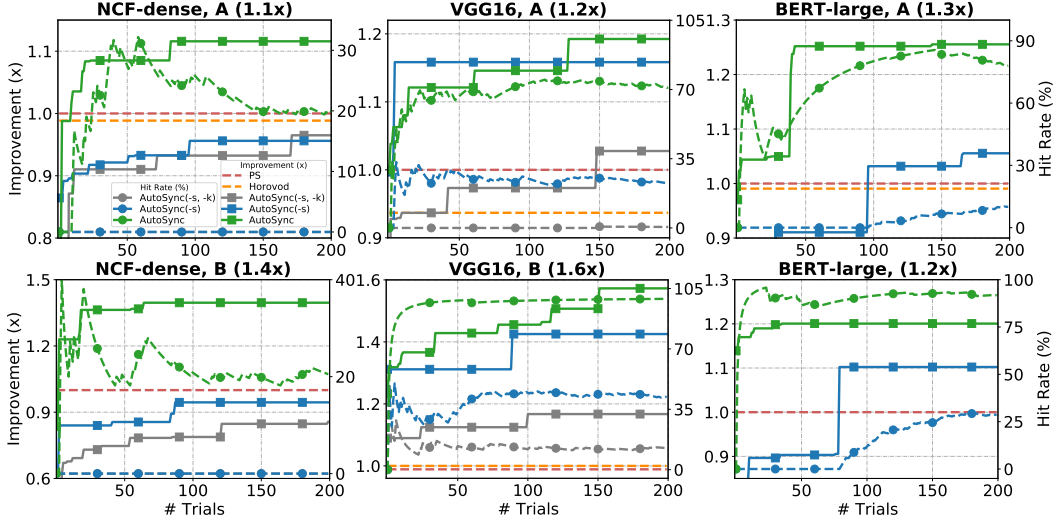

**Figure 2:** Comparing `AutoSync, AutoSync(-s), AutoSync(-s,-k)` on (left axis) the improvement (higher is better) of the best found strategy over baseline, (right axis) the percentage of strategies better than baseline (higher is better), w.r.t. the number of trials conducted in 200 trials. The baseline (1x) is the better one of `PS` and `Horovod`). A curve is skipped from the plot if it is too below the baseline. The average over 3 runs is reported, the std of hit rates are in $\pm 3\% - 7\%$ and that of improvements over baseline at trial 200 are in $\pm .04 - .1$.

1.25x faster than `PS` in about 50 trials, and 1.2x using only 30 trials on Cluster B. Similar patterns are observed in other settings as well.

**Search space evaluation.** We define the metric *hit rate* – for each auto-optimization run, the hit rate at trial $k$ is calculated as percentage of strategies explored up to trial $k$ that are above hand-optimized baselines. We calculate the hit rates independently for each run and report the average percentage at $k$ of 3 runs in the right axis of Figure 2. Except on (NCF-dense, B), `AutoSync(-s)` reports positive hit rates and the rate is acceptable on (VGG16, B) (42.0%) and (BERT-large, B) (28.5%). When augmented with a simulator, `AutoSync` frequently hits better strategies, especially on more complex models VGG16 and BERT-large ($> 70\%$). This verifies that factoring the strategy w.r.t. each variable and co-optimizing multiple factors form a promising space with a profound set of strategies better than manually optimized.

**Knowledge constraints.** Figure 2 contrasts another variant, `AutoSync(-s,-k)`, where the knowledge constraints in §3.2 are removed from `AutoSync(-s)`. `AutoSync(-s,-k)` mostly visits strategies below hand-optimized baselines, especially on complex models with a larger search space – on BERT-large all strategies by `AutoSync(-s,-k)` are far below baselines hence are skipped in the plots. Incorporating system design principles as knowledge is key to make the search manageable.

**Feature importance.** We ablate $z^{raw}$ and $z^{pre}$ and reveal their individual effect in Table 3. Specifically, we train RNN simulators with: (1) only predefined features $z^{pre}$, (2) only raw features $z^{raw}$, (3) the full features, under 6 $(\mathcal{G}, \mathcal{D})$ settings, and compare their ranking accuracy on test sets. Using full features achieves the best accuracy, demonstrating that the predefined and raw features may contain complementary information. On the other hand, the predefined features can be beneficial at the initial phase of search as it is possible to directly rank strategies using them without training.

| Setting | Full | Raw only | Predefined only |
|---|---|---|---|
| NCF-dense, A | **0.894** | **0.894** | 0.883 |
| NCF-dense, B | **0.913** | 0.907 | 0.873 |
| VGG16, A | **0.796** | 0.785 | 0.734 |
| VGG16, B | **0.839** | 0.837 | 0.816 |
| BERT-large, A | 0.847 | 0.848 | **0.850** |
| BERT-large, B | **0.755** | 0.746 | 0.735 |

**Table 3:** Studies on feature importance (pairwise ranking accuracy is reported). The std is within $\pm 2\%$ (3 runs).

## 5.2 Transferring Trained Simulators

**Transferability studies.** Different from §5.1, we train RNN simulators using trial data from a source domain $\mathcal{G}_s, \mathcal{D}_s$ to rank the strategies in unseen target domains $\mathcal{G}_t, \mathcal{D}_t$, without any additional training, and report the ranking accuracy in Table 4. In general, we note that: (1) Models with

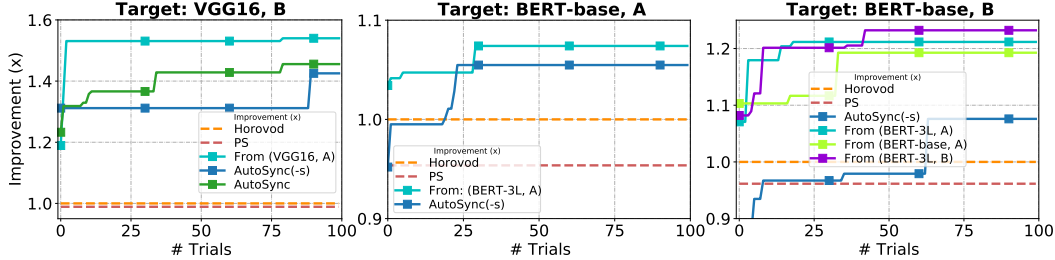

**Figure 3:** Transferring trained simulators from different source domains to 3 target domains, compared to untransferred `AutoSync-` and `AutoSync` with a budget of 100 trials. The average of three runs is reported.

similar architectures exhibit higher transferability. With fixed $\mathcal{D}$, transformer [31] based models transfer between each other pretty well, with the lowest accuracy at $0.76$; the transferability is slightly compromised when $\mathcal{D}$ is changed, due to increased domain distance. This hints we can use a simulator trained for smaller BERT models to optimize the strategy of a larger BERT model. (2) Models with similar distributions of the size of variables demonstrate transferability. For instances, the NCF-dense model, though only with 10 variables, can transfer to BERT-base with a decent accuracy $0.7904$; VGG-16, as a CNN, ranks strategies of BERT-base better ($0.7298$) than that of ResNet101 ($0.56$); both VGG16 and BERT-base have uneven communication loads caused by several extremely large variables. (3) When $\mathcal{G}$ is fixed, transferability is observed across $\mathcal{D}$. In practice, we might pretrain a simulator using in-house cheap clusters (e.g. cluster A), and deploy the simulator for training jobs on more expensive and larger-scale clusters.

**End-to-end results on transfer learning.** We now transfer trained simulators to guide end-to-end optimizations. We deliberately target difficult-to-parallelize models and expensive clusters: (BERT-base, A), (BERT-base, B), and (VGG16, B). We set a smaller budget of 100 trials, and *do not use* any data from target domains for model selection or additional finetuning. Figure 3 illustrates the optimization progress. On (VGG16, B), surprisingly, a well-trained simulator from (VGG16, A) can find strategies 1.5x faster as soon as in 5 trials. On (BERT-base, B), we experiment with 3 source domains and notice that the domain distance might impact the end-to-end results: source (BERT-3L, B) achieves better quality and efficiency than (BERT-base, A). We defer a more thorough study of the transferability to future studies. Overall, we manage to transfer simulators trained on "cheap" domains to find good strategies in only a few trials in "expensive" domains. Besides the reduction on the number of trials, transferring a simulator effectively decreases the wall-clock time taken in auto-optimization as it bypasses simulator training, which is advantageous for scenarios where single-shot model training happens often.

| Source → Target | Accuracy |
|---|---|
| (BERT-3L, A) → (BERT-3L, B) | 0.8672 |
| (Transformer, A) → (BERT-base, A) | 0.7674 |
| (BERT-3L, A) → (BERT-base, A) | 0.7591 |
| (BERT-3L, B) → (BERT-base, B) | 0.8100 |
| (NCF-dense, A) → (BERT-base, B) | 0.7904 |
| (VGG16, A) → (BERT-base, B) | 0.7298 |
| (BERT-3L, A) → (BERT-base, B) | 0.6992 |
| (BERT-base, A) → (BERT-base, B) | 0.6852 |
| (VGG16, B) → (BERT-base, B) | 0.6774 |
| (Transformer, A) → (BERT-base, B) | 0.6672 |
| (BERT-3L, A) → (Transformer, A) | 0.8866 |
| (Transformer, B) → (Transformer, A) | 0.8305 |
| (Transformer, A) → (Transformer, B) | 0.8171 |
| (BERT-3L, A) → (Transformer, B) | 0.808 |
| (NCF-dense, A) → (NCF-dense, B) | 0.7694 |
| (VGG16, A) → (VGG16, B) | 0.7966 |
| (BERT-3L, A) → (VGG16, B) | 0.5583 |
| (ResNet50, A) → (ResNet101, A) | 0.6057 |
| (VGG16, A) → (ResNet101, A) | 0.5600 |
| (VGG16, A) → (ResNet50, A) | 0.7156 |
| (VGG16, A) → (DenseNet121, A) | 0.7596 |
| (ResNet101, A) → (ResNet101, B) | 0.7187 |
| (DenseNet121, A) → (ResNet50, B) | 0.5266 |
| (VGG16, A) → (InceptionV3, A) | 0.7857 |

**Table 4:** The target domain test accuracy under different transfer learning settings.

## 6  Conclusion

The proposed AutoSync constructs a novel search space of synchronization-affecting factors and learns to optimize synchronization strategies on variable-level towards better training performance. When guided by an ML-based simulator and proper prior knowledge, AutoSync is able to find synchronization strategies up to 1.6x better than those manually optimized, even with only 200 trial run training data for the simulator. A dataset accompanying the proposed framework containing 10000 strategy, model, cluster, and runtime tuples will be made available to facilitate future research in automating the parallelization of ML programs.

## Broader Impact

The proposed AutoSync alleviates the burden on ML researchers and practitioners in choosing appropriate synchronization strategy for efficient distributed training, enables substantial speed up of ML prototyping and training, and reduces the cost of their operational workloads using distributed computing. Further, AutoSync is transferable to unseen model and cluster settings by the design of domain-agnostic features. By this, finding a good synchronization strategy for a large-scale ML model such as BERT [7] and GPT [28] or on a relatively expensive cluster only requires developing runtime simulators using data collected from a streamlined model on handy clusters, saving substantial experimental efforts and budgets. We will release and open-source our code and a new dataset to benefit the research community, to democratize high-performance ML systems, and make them accessible to non-ML-educated software developers and society at large. Since such needs are prevalent across many disciplines beyond computing and information science – such as industrial and manufacturing, healthcare, biology, social science, and finance – our deliverables are expected to have a catalytic impact.

## Acknowledgement

Hao Zhang, Xiaodan Liang, and Eric P. Xing are sponsored by Petuum, Inc. The work was done during Yuan Li's internship at Petuum, Inc. Yuan Li and Lawrence Carin are supported by the Office of Naval Research under grant N00014-18-1-2871. Zhijie Deng is supported by the National Key Research and Development Program of China (No.2017YFA0700904), NSFC Projects (Nos. 61620106010, U19B2034, U1811461).

## Footnotes

[1]The data and code accompanying this paper are available at https://github.com/petuum/autodist.

[2]Since we focus on synchronous training and optimizing system throughput, we exclude optimizations beyond data-parallel training (*e.g.,* operation partitioning in model-parallel training), or introduce deviation of parameter updates (*e.g.,* staleness).

[3]In this paper, we do not consider synchronization-affecting factors that would alter the algorithm or result as in the original single-node code, such as lossy compression [21], staleness [14], etc.

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
