[Supplementary Material]

# Supplementary Material
# AutoSync: Learning to Synchronize for Data-Parallel Distributed Deep Learning

[1,2]**Hao Zhang**[†], [1,3]**Yuan Li**[†], [4]**Zhijie Deng**, [1]**Xiaodan Liang**,
[3]**Lawrence Carin**, [1,2]**Eric P. Xing**
[1]Petuum Inc., [2]Carnegie Mellon University, [3]Duke University, [4]Tsinghua University
{hao.zhang,christy.li,xiaodan.liang,eric.xing}@petuum.com,
dzj17@mails.tsinghua.edu.cn, lcarin@duke.edu

The supplementary material is organized as follows:

- §1 explains the rationale of estimating $T$ using $T = \max(T_{\text{comp}}, T_{\text{sync}})$ (Section 3.1 of the main paper).

- §2 provides more detailed explanations about the predefined modeling of $T_{server}^{PS}$ and $z^{PS}$ in addition to the contents in Section 3.1 of the main paper.

- §3 provides details on the predefined modeling for $T^{CC}$ and the extraction of domain-agnostic features $z^{CC}$ (corresponding to Section 3.1 of the main paper).

- §4 gives detailed formulations of the RNN simulator (Section 3.1 of the main paper).

- §5 gives detailed formulations of the GAT simulator (Section 3.1 of the main paper).

- §6 provides detailed algorithms for the two knowledge constraints $c_{lb}$ and $c_{am}$ (Section 3.2 of the main paper).

- §7 elaborates the method for selecting the final set of candidate strategies for trial execution, and the approaches to estimates similarities between strategies (Section 3.2 of the main paper).

- §8 gives details about the AutoSync system implementations and baseline systems information (Section 5 of the main paper).

- §9 lists all the 11 models ($\mathcal{G}$) and 14 cluster specifications ($\mathcal{D}$) we have experimented with throughout our research, and provides information about the public dataset we are going to release. (Section 5 of the main paper).

- §10 provides some hyperparameter information about the simulator training.

## 1   Rationale of $T = \max(T_{\textbf{comp}}, T_{\textbf{sync}})$

In the main paper, we model the per-iteration runtime $T$ of parallelizing $\mathcal{G}$ on $\mathcal{D}$ using two contributing components: computation time $T_{\text{comp}}$, and parameter synchronization time $T_{\text{sync}}$, and obtain the per-iteration time via:

$$T = \max(T_{\text{comp}}, T_{\text{sync}}). \tag{1}$$

The rationale behind Eq. 1 is as follows: (1) Since many runtime systems (e.g. TensorFlow [1] or PyTorch [3]) introduce scheduling or parallelization between communication and computation, in practice, there are significant overlaps between the two components; (2) in data-parallel training, it is commonly observed that one component usually dominates the other [4]. These make using the maximum of them as the estimation of the total time reasonable.

## 2 More Explanations for $T_{server}^{PS}$

For modeling $T_{sync}$ of variables, we have introduced two hyperparameters: network overhead denoted as $\phi$, and GPU kernel memory latency denoted as $\delta$ which includes device-host transfer time and GPU kernel overhead. As $\delta$ is relatively slow compared with other time consumption, we treat it as a constant that does not scale with variable size. Synchronizing $v_i$ via PS ($v_i \in V^{PS}$) involves: (1) workers send gradients to servers, (2) servers update parameters, (3) servers send the updated parameters to workers, where (2) is negotiable and (1)(3) are symmetric processes and cost the same amount of time. We denote the original size (e.g. byte size) of the gradient of the variable $v_i$ as $m_i$ (note $m_i$ takes into consideration the sparsity of the gradients when applicable), and assume it will apply the encoding/decoding scheme $c_i \in s_i$, so the actual size of the message (related to $v_i$) to be transferred across devices is $c_i(m_i)$, where we also use $c_i$ to denote the compression function that reduces the original size $m_i$ to $c_i(m_i)$.

Let $w_i$ denote the number of workers involved in synchronizing $v_i$. The parameter transfer process involves transferring data between the GPU device memory and the host memory (RAM) within the same machine, and between the host memory across machines. The first process introduces GPU kernel latency and device-host communication. The second process introduces network overhead (*e.g.,* latency) and network communication. Hence, the communication time $T_{server}^{PS}$ on the server hosting $v_i$, indexed as $j$, is

$$T_{server}^{PS}(v_i) = \underbrace{\sum_{k=1}^{w_i} \mathbb{I}_d(j,k) \cdot \frac{c_i(m_i)}{b_{j,k}} \cdot r_{i,k}^{\mathbb{I}_p}}_{\text{network transfer}} + \underbrace{\sum_{k=1}^{w_i} \mathbb{I}_d(j,k) \cdot r_{i,k}^{\mathbb{I}_p} \cdot \phi}_{\text{network overhead}} + \underbrace{\delta}_{\text{GPU kernel latency}}, \quad (2)$$

where $r_{i,k}$ is the number of replicas of $\mathcal{G}$ on worker $k$ if the worker $k$ has multiple GPUs that host multiple replicas of $\mathcal{G}$ respectively; $\mathbb{I}_d(j,k)$ and $\mathbb{I}_p$ are true when server $j$ and worker $k$ locate on different machines and when hierarchical reduction is used, respectively. To interpret Eq. 2, the first term corresponds to sending messages from each worker $k$ to the server $j$ (and vise versa). The second term captures network overheads that scale linearly with the number of workers, or with the number of replicas when $\mathbb{I}_p$ is false. The third term captures the constant GPU memcpy latency.

The estimation $T_{server}^{PS}(v_i)$ can be accumulated across all $v_i \in V^{PS}$ to obtain the total synchronization time taken for all PS variables. We also derive the domain-agnostic features $z^{PS}$ based on Eq. 2, which has been elaborated in the main paper.

## 3 Predefined Modeling for $T_{sync}^{CC}$ and Domain-agnostic Features $z^{CC}$

In the main paper and §2, we have demonstrated the predefined modeling for communicating variables which adopt the PS communication architecture ($v_i \in V^{PS}$). Next, we develop the model for variables using collective communication (CC).

For $v_i \in V^{CC}$, we model 5 mostly used collective primitives: `AllReduce`, `ReduceScatter`, `AllGather`, `Broadcast` and `Reduce` [12]. Take an example when there are $w$ workers and the device order in the substrategy $s_i$ is a ring [14]. Each primitive sends and receives $\frac{2(w-1)}{w}$, $\frac{w-1}{w}$, $\frac{w-1}{w}$, 1, 1 times, respectively, in its applicable scenario for parameter synchronization (e.g. `AllReduce` for dense gradients or `AllGather` for sparse gradients [14]) . Therefore, we can obtain $T_{sync}^{CC}(v_i)$ using the following formula:

$$T_{sync}^{CC}(v_i) = \underbrace{\mathbb{I}_1 \frac{2(w_i-1)c_i(m_i)}{w_i b_m} + \mathbb{I}_2 \frac{(w_i-1)c_i(m_i)}{w_i b_m} + \mathbb{I}_3 \frac{c_i(m_i)}{b_m}}_{\text{network transfer}} + w_i \cdot \phi + \delta, \quad (3)$$

where $b_m = \min_{(k_1,k_2)\in \text{ring}} b_{k_1,k_2}$ denotes the lowest bandwidth between devices in the ring, since the throughput of a ring is restricted by the lowest bandwidth in the network [19]. $\mathbb{I}_1, \mathbb{I}_2, \mathbb{I}_3$ are true when `AllReduce`, `ReduceScatter` and `AllGather`, `Broadcast` and `Reduce` are activated, respectively. The formula is derived based on counting how many times each message (i.e. gradients) needs to be passed across the ring, taking into considerations both network transfer overhead as well as the device-host memory swap latency. The total synchronization time for variables assigned with collective communication is $T_{sync}^{CC}(V^{CC}) = \sum_{i=1}^{|V^{CC}|} T_{sync}^{CC}(v_i)$.

In a similar way as in §2, the domain-agnostic features for $v_i \in V^{CC}$ are $z_i^{CC} =$ [network transfer, coefficient of $\phi$, coefficient of $\delta$] from Eq. 3, and the global features of $T_{sync}^{CC}$ are $z^{CC} = \max\{z_i^{CC} \text{ for } v_i \in V^{CC}\} \oplus \sum\{z_i^{CC} \text{ for } v_i \in V^{CC}\}/|V^{CC}|$.

# 4  Detailed Formulation for the RNN-based Simulator

The formulation of applying RNN with both predefined and raw features on runtime prediction can be written as

$$z_i = \text{MLP}_i\left(z_i^{raw} \oplus z_i^{pre}\right), \tag{4}$$

$$h_{i+1} = \text{LSTM}\left(h_i, z_i\right), \tag{5}$$

$$h'_{i+1} = \text{LSTM}\left(h'_i, z_{|V'_{\mathcal{G},\theta}|+1-i}\right), \tag{6}$$

$$\hat{f} = \text{MLP}_o([h_{|V'_{\mathcal{G},\theta}|}, h'_{|V'_{\mathcal{G},\theta}|}]), \tag{7}$$

where $\text{MLP}_i$ converts input features to a hidden representation, and $\text{MLP}_o$ converts encoded features after LSTM fusion to the output logits. Note that the LSTM walks through each $v_i \in V'_{\mathcal{G},\theta}$ strictly following their original forward (backward) order in the computational graph, so as to inject this information into the modeling. Note the RNN works on the transformed graph $\{V'_{\mathcal{G},\theta}, E'_{\mathcal{G}}\}$, which additionally has partitioned variables compared to the original graph $\{V_{\mathcal{G},\theta}, E_{\mathcal{G}}\}$.

# 5  Detailed Formulation for the GAT-based Simulator

The formulation of applying GAT with both predefined and raw features on runtime estimation can be written as

$$\{z_i\}_{i=1}^{|V'_{\mathcal{G},\theta}|} = \text{GAT}(\{z_i^{raw} \oplus z_i^{pre}, \text{ for } v_i \in V'_{\mathcal{G},\theta}\}, E'_{\mathcal{G}}), \tag{8}$$

$$z = \frac{1}{|V'_{\mathcal{G},\theta}|} \sum(\{z_i\}_{i=1}^{V'_{\mathcal{G},\theta}}), \tag{9}$$

$$\hat{f} = q(z), \tag{10}$$

where GAT denotes the graph attention operations [18] applied on the graph $\{V'_{\mathcal{G},\theta}, E'_{\mathcal{G}}\}$ with $z_i^{raw} \oplus z_i^{pre}$ as node features. $q$ is a linear layer that converts hidden representation to runtime estimation.

# 6  Details of the Knowledge Constraints $c_{lb}$ and $c_{am}$

The two knowledge constraints, load balancing constraint $c_{lb}$ and adjacent merging constraint $c_{am}$, are implemented together with the strategy sampler as regulations. In particular, the strategy sampler first generates a partial strategy $\mathcal{S}$, leaving two fields, *variable placement* (PS-based) and the *group* (collective-based) undecided. A global load balancer ($c_{lb}$) and group assigner ($c_{am}$) assign their values using randomized and approximate solutions, illustrated in Algorithm 1 and Algorithm 2, respectively. Note both constraints work on the set of variables $V'_{\mathcal{G},\theta}$ with partitioned shards, instead of the original set of variables $V_{\mathcal{G},\theta}$.

The two constraints help filter strategies that are obviously inefficient, but approximately maintain global load balance on the cluster and adjacent merging structures correlated with the forward-backward propagation order. Their effectiveness has been validated in §5.1 in the main paper.

# 7  Select Final Candidate Strategies for Trial Execution

This section provides details on the selection of the final set of strategy candidates for trial execution.

As stated in §3.1 of the main paper, we minimize a weighted sum of $\hat{f}$ and internal similarity to select the final set of strategy candidates for trial execution. Formally, we solve the following optimization

---

**Algorithm 1:** Implementation of the load balance constraint $c_{lb}$

---

**Inputs:** $\mathcal{S}, \mathcal{D}, V'_{\mathcal{G},\theta}$

1 **Function** `load_balance_constraint`($\mathcal{S}, \mathcal{D}, V'_{\mathcal{G},\theta}$):
2     Set loads of each device in $\mathcal{D}$ as $l_{cur} = [0, ..., 0]$
3     Sum up the bandwidth of devices in $\mathcal{D}$: $b_{all} = \sum_{i=1}^{|\mathcal{D}|} b_{i,i}$
4     Sort $V'_{\mathcal{G},\theta}$ by byte size in descending order and get $V^{s'}_{\mathcal{G},\theta}$
5     **for** $v$ *in* $V^{s'}_{\mathcal{G},\theta}$ **do**
6         $l_{all} = \sum_{i=1}^{|\mathcal{D}|} l_{cur}[i]$
7         $logits = l_{all} * [b_{1,1}/b_{all}, \ldots, b_{|V_{\mathcal{D}}|,|V_{\mathcal{D}}|}/b_{all}] - l_{cur}$
8         $j \sim \text{Categorical}(\text{softmax}(logits))$
9         Set the placement of $v$ in $\mathcal{S}$ as $d_j \in \mathcal{D}$
10         $l_{cur}[j] = l_{cur}[j] + \text{byte\_size}(v)$

---

**Algorithm 2:** Implementation of the adjacent merge constraint $c_{am}$

---

**Inputs:** $\mathcal{D}, V'_{\mathcal{G},\theta}, \mathcal{G}$, the number of total collective merging groups $N$

1 **Function** `adjacent_merge_constraint`($\mathcal{D}, V'_{\mathcal{G},\theta}, \mathcal{S}, N$):
2     Sort $V'_{\mathcal{G},\theta}$ based on their location in the backpropagation order indicated by $\mathcal{G}$, from input to
      loss function, and get $V^{s'}_{\mathcal{G},\theta}$
3     Set loads $l_{cur} = [0, ..., 0]_{i=1}^{N}$ corresponding to $N$ merge groups
4     $l_{avg} = \frac{1}{N} \sum_{v \in V^{s'}_{\mathcal{G},\theta}} \text{byte\_size}(v)$
5     $g = 0$
6     **for** $v$ *in* $V^{s'}_{\mathcal{G},\theta}$ **do**
7         **if** $l_{cur}[g] \geq l_{avg}$ **then**
8             $g = g + 1$
9         **if** $l_{cur}[g] < l_{avg}$ **then**
10           Set the group of $v$ in $\mathcal{S}$ as $g$
11           $l_{cur}[g] = l_{cur}[g] + \text{byte\_size}(v)$

---

problem:

$$\min_{\{\mathcal{S}_i\}_{i=1}^{K} \subset \{\mathcal{S}_i\}_{i=1}^{M}} \sum_{i=1}^{K} \hat{f}(\mathcal{S}_i) + \alpha \sum_{i=1}^{K} \sum_{j=1}^{K} \text{sim}(\mathcal{S}_i, \mathcal{S}_j), \tag{11}$$

where $\{\mathcal{S}_i\}_{i=1}^{M}$ are the $M$ qualified strategies filtered first using the simulator score, $\alpha$ denotes a trade-off coefficient, and sim is a pairwise similarity function between strategies. Solving Eq. 11 helps deliver a set of candidates that trade off between low predicted runtime (exploitation) and low similarity (exploration). The above problem is a typical problem of submodular minimization, and we resort to the greedy algorithm for an approximate solution [10].

**Estimating the similarity between strategies.** Since we do not have a continuous representation of strategies, we have developed two approaches to estimate the similarities between strategies. The first approach estimates the similarity of the two strategies $\mathcal{S}_1, \mathcal{S}_2$ by comparing each sub strategy $s_{i1} \in \mathcal{S}_1$ and $s_{i2} \in \mathcal{S}_1$ corresponding to the variable $v_i$, and counting how many choices of each synchronization-affecting factor are the same, and use the final count as a measure of their similarity (higher is more similar). The second approach takes the cosine similarity between the hidden outputs of the simulator, whose inputs are $\mathcal{S}_1$ and $\mathcal{S}_2$, as a proxy of the strategy similarity. Empirically, we find the two approaches perform similarly, but the first similarity function does not depend on the simulator thus can be used when no trained simulator is available (e.g. for the baseline `AutoSync(-s)`).

# 8 More Details on AutoSync Implementations and Baseline Systems

In this section, we provide more details about the experimental setup, including the implementations of AutoSync backend for strategy generation and application, and the details about the two baseline systems: `Horovod` and PS.

**AutoSync System Implementation.** Frameworks such as TensorFlow offer a flexible distributed runtime [1, 20] for evaluating dataflow graphs on distributed clusters [1, 20]. On top of TF 2.0, we have built a composable dataflow graph rewriting system, called *AutoDist* (`https://github.com/petuum/autodist`), that allows composing different synchronization aspects, such as synchronizer, placement and partitioning of variables, etc. as synchronization strategies. More precisely, given the single-node dataflow graph $\mathcal{G}$ extracted from user code, AutoDist can rewrite $\mathcal{G}$ by modifying nodes, edges and their attributes, into a distributed graph $\mathcal{G}'$ with all required semantics from $\mathcal{S}$. It then launches TensorFlow distributed runtime to execute $\mathcal{G}$ on the cluster of nodes from $\mathcal{R}$. This allows evaluating different $\mathcal{S}$ using a common distributed engine and eliminating performance variations caused by different runtime. We skip the detailed descriptions about how AutoDist is designed as they are out of the scope of this paper.

Based on AutoDist, AutoSync is implemented a synchronization strategy auto-optimizer component therein. The specific code of AutoSync can be found at `https://github.com/petuum/autodist/tree/master/autodist`.

**System Setup.** All the baselines and AutoSync rely on TensorFlow 2.0 as the distributed runtime. For the experiment results reported in this paper, we compile TF2.0 with CUDA10.0, CUDNN 7.1, and NCCL 2.4.7, and uses gRPC for network communication.

**Baseline Systems.** In the main paper we introduce two strong hand-optimized synchronization systems as external baselines. We provide more details about the two systems.

`Horovod` [14] is one of the most adopted open source synchronization systems for data-parallel distributed ML. It uses *AllReduce* (*AllGather*) to synchronize dense (sparse) gradients of all model variables, and utilizes BO to autotune the merge scheme for multiple collective operations based on collected trial data in warm-up runs. Details of the optimization can be found at `https://github.com/horovod/horovod/blob/master/docs/autotune.rst`. Per our experiments on our cluster setup, it reports up to 2x speedup than a Google-provided parameter server implementation based on distributed TensorFlow as runtime (details at `https://github.com/tensorflow/examples/blob/master/community/en/docs/deploy/distributed.md`), on multiple CNNs such as ResNet101 and InceptionV3. `Horovod` uses NCCL 2.4.7 for collective communication, which is the same with AutoSync.

PS is a highly tuned parameter server implementation with multiple optimizations from recent PS literature incorporated. We elaborate a few notable optimizations: (1) it maintains load balance by partitioning large variables and evenly placing the shards across servers depending on their available bandwidth (i.e. correlates to its maximum bandwidth and the current load as a parameter server); (2) it uses the BO algorithm to decide the partitioning size following ByteScheduler [13]; (3) it communicates sparse gradients (*IndexSlices* in TensorFlow) using *Gather* and *Scatter* primitives (instead of *Reduce* and Broadcast) [9] to reduce communication overhead; (4) it performs hierarchical Reduce (or Gather) and Broadcast (or Scatter) on nodes with more than 1 GPUs to reduce network traffic. The optimizations are implemented on top of TensorFlow 2.0 as well, so the backend also relies on distributed TensorFlow 2.0 for distributed execution, same with AutoSync.

# 9 Descriptions of the Experiment Setup and Public Dataset

## 9.1 Model and Cluster Specifications

Throughout our research in this paper, we have conducted experiments on 10 models from different DL families, and 14 cluster configurations based on our in-house cluster and AWS. We provide additional details about the models and cluster specifications below and in Table 1 and Table 2.

**Models.** Table 1 list the details of the 11 models we have experimented with, including 5 CNN models for image classification (IC), a transformer-based model for neural machine translation (MT), two versions of the neural collaborative filtering model for recommendation systems, and four BERT

| Model | Task | Training data | bs/gpu | #params |
|---|---|---|---|---|
| ResNet50 [6] | IC | ImageNet | 32 | 23M |
| ResNet101 [6] | IC | ImageNet | 32 | 45M |
| InceptionV3 [16] | IC | ImageNet | 32 | 24M |
| VGG16 [15] | IC | ImageNet | 32 | 138M |
| DenseNet121 [8] | IC | ImageNet | 32 | 8M |
| Transformer [17] | MT | WMT'14 ende | 5K | 62M |
| NCF-dense [7] | CF | MovieLens-20Mx16x32 | 256 | 122M |
| NCF-sparse [7] | CF | MovieLens-20Mx16x32 | 256 | 122M |
| BERT-3L [2] | LM | Wiki & BookCorpus | 32 | 11M |
| BERT-6L [2] | LM | Wiki & BookCorpus | 32 | 36M |
| BERT-base [2] | LM | Wiki & BookCorpus | 32 | 110M |
| BERT-large [2] | LM | Wiki & BookCorpus | 8 | 340M |

Table 1: Models we have experimented with. Their implementations are from the `tensorflow/models` repository. IC: image classification, MT: machine translation, LM: language modeling.

variants for language modeling. Their single-node implementations are from the official open source repository at https://github.com/tensorflow/models. Their training data, and number of parameters are listed in Table 1. We use default training settings suggested by MLPerf [11]. The per-GPU batch size is indicated in Table 1 as well.

It is worth noting that, for the neural collaborative filtering (NCF) model, we follow MLPerf [11] and use an enlarged version (x16x32) of the model and training set – which expands the original MovieLens-20M dataset [5] with 16x more users and 32x more movies. Details can be found at https://github.com/mlperf/training/tree/master/recommendation/pytorch#getting-the-expanded-dataset. To test the system's capability, we also experiment with two versions of NCF: NCF-dense, which uses dense tensors to represent the gradients of embedding variables in the model, and NCF-sparse, which uses sparse tensors (*IndexSlices* in TensorFlow) to represent the gradients instead.

**Clusters.** We focus on GPU clusters since it is the main setup for distributed DL training. During our research, we have tested on 14 different cluster configurations as listed in Table 2: *Cluster A* includes maximally 16 nodes, each equipped with a GeForce TITAN X GPU, an Intel 16-core CPU and 64GB RAM, interconnected via a 40-Gigabit Ethernet switch; *Cluster B*, based on AWS, consists of up to 8 node, each node is one of the g3.4xlarge (1x Tesla M60 GPU and 10GbE Ethernet), g3.16xlarge (4x M60, 25GbE), g4dn.2xlarge (1x NVIDIA T4 GPU and 25GbE Ethernet), g4dn.12xlarge (4x T4, 50GbE) instance types. Details about the configurations of AWS instances can be found at https://aws.amazon.com/ec2/instance-types/g4/ and https://aws.amazon.com/ec2/instance-types/g3/. Due to AWS constraints, they all have 10GbE single-flow bandwidth. On top of these two clusters, we list all the resource specifications we have experimented with, including cluster setup, number of GPUs on each cluster node, and bandwidth information, in Table 2.

To avoid confusion, it is worth noting that the setup A2 and B7 in Table 2 correspond to the `Cluster A` and `Cluster B` referred to in the main paper.

## 9.2 Public Datasets

We have collected a dataset containing nearly 10k data points of $\{(\mathcal{G}, \mathcal{D}, \mathcal{S}), f\}$, where $\mathcal{G}$ is one of the DL models in Table 1, $\mathcal{D}$ is one of the cluster setups from Table 2, $\mathcal{S}$ is randomly sampled strategy from the proposed strategy space, and $f$ is the groundtruth runtime collected via real distributed execution.

The dataset contains strategies sampled for all 11 models, ranging from fixed-formed strategies such as those used in specialized systems, and randomly explored strategies by AutoSync during the strategy auto-optimization. The data are organized into multiple folders where each folder corresponds to a domain of $(\mathcal{G}, \mathcal{D})$. Hence the dataset used in the main paper is a subset containing several domains belonging to the entire collected dataset.

| $\mathcal{D}$ | Setup | GPU distribution | Bandwidth spec |
|---|---|---|---|
| A1 | 16x Cluster A nodes | [1] x 16 | 40GbE |
| A2 | 11x Cluster A nodes | [1] x 11 | 40GbE |
| B1 | 2x g3.16 | [4] x 2 | 25 GbE |
| B2 | 3x g3.16 | [4] x 3 | 25 GbE |
| B3 | 4x g3.16 | [4] x 4 | 25 GbE |
| B4 | 1x g4dn.12 | [4] x 1 | 50 GbE |
| B5 | 2x g4dn.12 | [4] x 2 | 50 GbE |
| B6 | 3x g4dn.12 | [4] x 3 | 50 GbE |
| B7 | 4x g4dn.12 | [4] x 4 | 50 GbE |
| B8 | 8x g4dn.12 | [4] x 8 | 50 GbE |
| B9 | 1x g3.4, 1x g3.16 | [1, 4] | 10/25 GbE |
| B10 | 1x g3.16, 1x g4dn.12 | [4] x 2 | 25/50 GbE |
| B11 | 2x g3.16, 2x g4dn.12 | [4] x 4 | 25/50 GbE |
| B12 | 1x g4dn.2, 1x g4dn.12 | [1, 4] | 25/50 GbE |

Table 2: Cluster specifications we have experimented with, listed with their reference name ($\mathcal{D}$), setup information (Setup), GPU distribution and bandwidth specifications (Bandwidth spec).

For quick access, we have provided scripts that read $\mathcal{G}$ as dataflow graphs in standard TensorFlow 2.0 format, and read the strategies and runtime into json formats. Instructions on how to access the dataset are provided at https://github.com/petuum/autodist.

## 10 Simulator Training Dataset and Hyperparameters

For the end-to-end results (Fig.2 in the main paper), we train simulators using runtime data collected on-the-fly during trials, strictly following the workflow illustrated in Fig.1 of the main paper.

For the ablation studies (Table.1-3), we split pre-collected data at auto-optimization of a specific domain (e.g., (NCF-dense, A)) into train/val/test at 70%/15%/15%, respectively, and report the ranking accuracy on the test split (averaged over 3 runs).

Training RNN simulators in all settings use Adam with 1e-3 lr, decayed by 0.3 at the 80th/160th epoch, for 200 epochs. We clipped the gradient norm to 0.25.