[Reviews · NeurIPS 2020]

Review 1

Summary and Contributions: This paper proposes a automatic algorithm to design synchronization strategies in data-parallel distributed training scenario. Taking many aspects which affect throughput of data-parallel training system into account, this paper use machine learning method to find optimal sync strategies, which achieves better result compared with hand-craft ones and the strategies also shows good transferability.

Strengths: This paper formulate sync strategy selection problem as a learning problem and solved with data-driven machine learning models to estimate the runtime of strategy proposals without launching real distributed execution. This is a valuable exploration, especially for industry. I think this paper is quite novel and the experimental evaluation is convincing.

Weaknesses: More analysis about linear model, RNN and GAT should be added, especially the reason why RNN perform best. I think it is not a natural way to treat z_i as a sequence. Experiments are only conducted on 2 fully-connected clusters, it would be better to evaluate on clusters with other topologies. --- My concerns are well addressed, and I will keep my score unchanged.

Correctness: To the best of my knowledge, the claims and method are correct.

Clarity: I have questions about the experiment. In section 5.1, ranking accuracies on test set were reported in Table 1. What is the test set here? Why is ranking accuracy used?

Relation to Prior Work: Besides aggregated gradient based data-parallel methods, there are also communication-efficient algorithms to improve efficiency data-parallel training. For example: EASGD[1], BMUF[2], which are of lower communication frequency for periodic sync was used. It would also be interesting to apply autosync to these kind of methods. [1] S. Zhang, A. Choromanska, Y. LeCun, Deep learning with Elastic Averaging SGD [2] K. Chen, Q. Huo, Scalable training of deep learning machines by incremental block training with intra-block parallel optimization and blockwise model-update filtering

Reproducibility: Yes

Additional Feedback:


Review 2

Summary and Contributions: This work present a method to optimize automatically the synchronization strategy for data-parallel distributed deep learning. It presents a way of representing the parallel training that is agnostic to hardware specifications together with a simulation method to cheaply estimate the computational cost due to hardware (ex: network communication). A transfer learning method is also proposed to leverage simulators fitted on different model architectures and cluster configurations.

Strengths: The proposed solution is complex with many moving parts and this work presents wide ablation study encompassing the effect of the model behind the simulation, the type of data used for the model, the simulation itself and knowledge constraints. This provides not only results to support the validity of the method but also insights on the role of the different parts. While the simulation already lowers the computational cost of the automated optimization of the synchronization strategy, the authors additionally propose a way of transferring the simulator to similar model architectures and cluster configuration. This enables even cheaper automated optimization.

Weaknesses: Automated optimization of synchronized strategies is a very specific problem, which limits the applicability of the solution. It remains however an relevant topic as synchronization is an important and expansive procedure of data-parallel distributed training. The black box optimization methods used for the optimization are fairly limited. Bayesian optimization would have been a good candidate, both because it is widely used for black box optimization and because it is used in Horovod for auto-tuning. Although I understand the difficulty to access many different clusters, having only 2 different ones makes it risky to claim generalization to widely different clusters.

Correctness: My main concern with the experiments is the small number of cluster configuration (2) and the lack of information on how results are reported. In figure 2, it says that the average over 3 runs is reported. The other runs should be plotted as well. The standard deviation could be added but for only 3 runs it is not very reliable. Likewise, the average is not very reliable with only 3 points unless the true standard deviation is very small, which I highly doubt. I find it surprising that the percentage of strategies that are better than manual tuning decreases in some case with additional trials. It is not clear how these rates are computed if not over the 3 runs, and the results suggest that it is indeed not the case otherwise we could only have 4 values (0/3, 1/3, 2/3, 3/3) in the plots. The same comment applies for all tables and figure 2-3. There is no information on the standard deviation. Beside these concerns, I highly appreciate the breadth of the ablation study.

Clarity: The problem formulation is well written. Being unfamiliar with this topic I had no difficulty following. Figure 1 caption should be improved. I understand there is a limit of space but the additional page of the camera-ready version should be used in part to describe Figure 1 in more details. Figure 2 needs to be improved, it is unreadable unless we make a big zoom-in. Hit-rate should be in a separate plot below, sharing the x axis only. It's hard to read.

Relation to Prior Work: I am not familiar with the domain of distributed training so I cannot assert if the authors covered the related literature properly. From what I can tell, the work is well compared with previous contributions in section 4.

Reproducibility: No

Additional Feedback: Section 3.1 Equation (2) I believe $p$ is missing in $r^{\Pi}_{i,k}$. Section 5.1. The example of 7 days and $2200 AWS credits saving should be given in the context of the full cost. 7 days saved out of how many for training, and $2200 AWS credits out of how many? In subsection 'Search space evaluation' I don't understand how 42% for VGG16 and 28.5% can be considered as a large positive hit rate. They way I understand it it means 58% and 71.5% of the strategies were worst than hand-optimized baselines. Why no hit-rate in Figure 3? Table 3 would be more informative in terms of improvement measures. It's hard to relate all the test accuracies. I don't understand why transferring the simulator would affect the performance. The model would be trained on the real data anyhow, isn't? The transferred simulation is used to chose the strategy, which should not impact much the final accuracy. # Post-Rebuttal I read the authors feedback and other reviews. I am satisfied by the authors answer and will accordingly increase my score to 7. The justification for BO based on the high dimensionality of the search-space is reasonable and should probably be included in the final version of the paper. I trust the authors to amend the paper with STD, more objective tone about the hit-rates and give context for the reduction of cost (saving $2200 out of $13561 is certainly significant).


Review 3

Summary and Contributions: A learning-to-synchronize algorithm, namely AutoSync, is proposed to automatically search the optimal synchronization strategy for deploying a distributed deep learning system. The authors systematically elaborate on this novel AutoML system from building the configuration space, optimizing the search algorithm, and evaluating the system performance. Experimental results on several benchmark datasets and common neural architectures were provided in terms of training time.

Strengths: 1) An efficient AutoSync system is provided, which could benefit the deep learning and AutoML community. 2) The configuration search space is clearly defined for a data-parallel distributed deep learning system, covering four different kinds of factors related to synchronization. 3) The proposed learning-to-synchronize algorithm is well formulated in a closed-loop optimization framework, which first initializes the system with graph-agnostic features and then iteratively refines the model with the learned graph-specific features. Moreover, the authors propose to use some knowledge constraints to further guide the searching process. 4) Extensive experimental results were provided to show the effectiveness of the proposed AutoSync over baselines.

Weaknesses: 1) The proposed AutoSync is trained upon the ground truth runtime, which, I suppose, should be dependent on a specific cluster system. What’s the transferability of the proposed AutoSync between different cluster platforms? For example, for the same graph, can the AutoSync model pre-trained on one platform be directly used on the others? 2) It is recommended to show more details about training the simulators, as well as the training set used in the experiment.

Correctness: Technically sonded.

Clarity: The paper is easy to follow.

Relation to Prior Work: The literature review is good to me.

Reproducibility: No

Additional Feedback: Post rebuttal comments: After reading the authors' responses, my main concerns have been well addressed. Thus, I keep my score unchanged.


Review 4

Summary and Contributions: AutoSync enables automatically search for synchronization strategies given a model structure and cluster configuration (domain). Using the learned strategy obtained from smaller domains, transfer learning can be used to derive strategies without additional training for larger ones. I still give a 6 after the author's response. Laking a strong baseline is still the main weakness of this paper in my opinion.

Strengths: The design of search space is novel. Instead of enumerating different possible combinations of model graphs G and cluster graphs D. The authors carefully decomposed the systems/strategies into orthogonal factors. It's an very interesting application of ML technique to ML systems. It might inspire more related works to apply ML to improve systems efficiency.

Weaknesses: With all due respects, I feel the baseline used by this paper is a bit weak. It might be useful to add experiments with settings identical to multi-replica tasks of MLPerf, and compare the results obtained from AutoSync and the (winning) MLPerf submissions. This work also introduced a learnable simulator capable of estimating the runtime of any domain (G, D). However, there is no direct validation or measurements on the runtime estimations.

Correctness: Claims made by this paper are objective in my opinion.

Clarity: This paper described the search space, runtime evaluation estimation, search algorithms and the transfer learning algorithms clearly. It might also help appreciation of this paper if some of the found strategy from AutoSync can be illustrated so readers can have a better understanding what AutoSync optimized.

Relation to Prior Work: The authors did a good job in literature review and covered the related works.

Reproducibility: Yes

Additional Feedback: I still give a 6 after the author's response. Laking a strong baseline is still the main weakness of this paper in my opinion.

[Author Response · NeurIPS 2020]

We thank all reviewers for the constructive feedback. We are encouraged by the acknowledgement of novelties of our

formulation [R1, R2, R3, R4], the comprehensiveness of our studies [R2, R3], and potential industry impact [R1, R2,

R4]. We address specific comments below and will incorporate them to the updated version.

[R1] **Performance analysis of 3 models; Modeling $z_i$ as a sequence is unnatural.** Many DL models contain only

few non-sequential *variable-to-variable* connections, e.g. only in embedding variables of BERT and NCF. Feeding $z_i$

into RNN following the data flows presented in $\mathcal{G}$ suffices for the RNN to capture the variable-to-variable structure.

For GAT, we observe large discrepancy between training and validation accuracy; it achieves lower-than-expected

performance due to the scarcity of training data and overfitting. Besides, the linear model cannot leverage raw features.

[R1, R2, R4] **Experiment w/ more clusters and baselines.** Throughout our study, we have tested on 14 different

clusters (Table.3 of the supplementary). *We managed to find better-than-hand-optimized strategies on 13 configurations*,

for the 3 models used in paper, except on B4 where we matched Horovod. A fair comparison to winning entries in

MLPerf is challenging due to the lack of access to similar setups (e.g., TPUs, P100-equipped clusters, their code).

We compare to SoTA PS implementation BytePS [27] (despite of a different distributed runtime) on (BERT-large, B):

AutoSync finds *1.3x faster* strategies in 200 trials. We'd be happy to submit an entry to MLPerf using our clusters.

[R1, R3] **Test set of Table.1 & details on simulator training.** For the end-to-end results (Fig.2), we train simulators

using runtime data collected on-the-fly during trials, strictly following Algorithm.1 in the supplementary. For the

ablation studies (Table.1-3), we split pre-collected data at auto-optimization of a specific domain (e.g., (NCF-dense,

A)) into train/val/test at 70%/15%/15%, respectively, and report the ranking accuracy on the test split (averaged over 3

runs). More information about the dataset is in §11.2 of the supplementary. Training RNN simulators in all settings use

Adam with 1e-3 lr, decayed by 0.3 at the 80th/160th epoch, for 200 epochs. We clipped the gradient norm to 0.25.

[R1, R3, R4] **Using ranking accuracy in Table.1; no measurement of real runtime estimation**: We train simulators

with an augmented ranking loss (see L187-194) instead of regression loss (to estimate runtime), as (1) auto-optimization

only requires predicting whether a strategy is better than others; (2) we internally tried predicting runtime and found it

resulting in worse end-to-end optimization performance; (3) ranking loss can transfer better across domains.

[R1] **Connection to EASGD and [Chen et al.].** EASGD can be thought of as an optimizer (equivalents are

Adam/Adagrad). So far, AutoSync optimizes system throughput without interfering with optimization results, and lets

users decide which optimizer. In future work, we can add an "optimizer" aspect (including EASGD) to the strategy

space, and extend Eq.1 to optimize model convergence. Similar reasoning applies to the technique in [Chen et al.].

[R2] **Considering Bayesian optimization (BO).** Applying BO for strategy optimization is theoretically feasible, but

faces two main obstacles: the strategy feature ($z^{pre} \oplus z^{raw}$) is very high-dimensional; Encoding variable-specific

choices of multiple synchronization aspects, it has variable-length. These prevent an easy adoption of BO in our case.

[R2] **Improvement on presentations, and standard deviation (std) in results**. Thanks for the suggestions! We

reinspect the results: the std of 3 runs for (1) ranking accuracy results (Table.1-3) are under $\pm 2\%$; (2) hit rates (Fig.2)

are in $\pm 3\% - 7\%$; (3) improvements over baseline at trial 200 (Fig.2) are in $\pm .04 - .1$. We'll extend results from 3 to 5

runs, add std in tables and plots, add hit rates in Fig.3, and fix presentation issues raised by R2.

[R2] **Computation of hit rates; decreased hit rates in some plots**. For each run, the hit rate at trial $k$ is calculated

as percentage of strategies explored up to trial $k$ that are above hand-optimized baselines. We calculate the hit rates

independently for each run and report the average percentage at $k$. The hit rates plateau or decrease majorly due to

randomness at exploration. 42%/28% hit rates for random exploration without a simulator (AutoSync(-s)) is higher

than our expectation; we'll adjust the tone to be more objective.

[R2] **Context of $2200 AWS credits saving.** Training BERT-large on Wikipedia+BookCorpus needs ≈2M iterations to

report results w/ bs = 128 [7, 23]. Suppose we rent cluster B ($3.912/h/node) and use the (maximally doable) per-GPU

bs = 8, training using the hand-tuned PS (Fig.2 bottom-right plot, 1.56s/iter) will spend $2M * 1.56 * 4 * 3.912/3600 =$

$13561.6. AutoSync takes 50 trials (2.5K iterations) to find a 1.2x faster strategy than PS. Hence, the net saving is

approximated as: $13561.6 - 2M * 1.56/1.2 * 4 * 3.912/3600 - 2.5K * 1.56 * 3.912 * 4/3600 \approx \$2243$, per job.

[R1, R2] **Measures and implication of accuracy results in Table.3**. Sorry for the confusion. To clarify, the test

accuracy in Table.3 is obtained by transferring a simulator trained on source domain data to infer target domain data

*without any additional training*. Simulators with closer accuracy to those in Table.1 and 2, which are trained using target

domain data, are more competitive to guide the auto-optimization (normally >70% suffices to guide the exploration).

[R3] **Transferability between different clusters**. Due to the feature design (see §3.3), the simulators do transfer

between cluster platforms (see §5.3). Table.3 and Fig.3 show proven results (in both ranking accuracy and end-to-end

results) that the simulator can transfer between cluster A and B with different setups *without additional training*.

[R4] **Visualizing strategies**. We inspected strategies over 20 $(\mathcal{G}, \mathcal{D})$ settings and found: (1) PS is dominant in dense

BERT-large; other settings have mixed PS/collective. (2) Inconsistent with [18], sparse variables on cluster B prefer

being synchronized via AllGather than PS on many models. (3) "Partition-then-allreduce" patterns (contrast to "merge-

then-allreduce" pattern in existing systems) are observed in BERT-large and NCF-dense on cluster B. We'll add a

section covering qualitative analysis of these strategies.

[Meta-Review · NeurIPS 2020]

The authors cast the task of parallel training as a learning problem, allowing data driven decisions to be made instead of the hand-crafted rules. The topic is relevant and the results are impactful. The comprehensive ablation studies performed to evaluate the system are also appreciated. Several aspects of the proposed system have room for improvement, both in terms of scope and quality. However, that doesn’t seem to be a crucial problem with the paper but rather room for follow up works.